# Spatial Distribution of *Pinus koraiensis* Trees and Community-Level Spatial Associations in Broad-Leaved Korean Pine Mixed Forests in Northeastern China

**DOI:** 10.3390/plants12162906

**Published:** 2023-08-09

**Authors:** Unil Pak, Qingxi Guo, Zhili Liu, Xugao Wang, Yankun Liu, Guangze Jin

**Affiliations:** 1Center for Ecological Research, Northeast Forestry University, Harbin 150040, China; pustpui201910@163.com (U.P.); gqx1965@126.com (Q.G.); liuzl2093@126.com (Z.L.); 2Key Laboratory of Sustainable Forest Ecosystem Management-Ministry of Education, Northeast Forestry University, Harbin 150040, China; 3Northeast Asia Biodiversity Research Center, Northeast Forestry University, Harbin 150040, China; 4CAS Key Laboratory of Forest Ecology and Management, Institute of Applied Ecology, Chinese Academy of Sciences, Shenyang 110164, China; wangxg@iae.ac.cn; 5Heilongjiang Forestry Engineering and Environment Institute, Harbin 150040, China; liuyankun1979@126.com; 6Key Laboratory of Forest Ecology and Forestry Ecological Engineering of Heilongjiang Province, Harbin 150040, China

**Keywords:** spatial point pattern, Korean pine, pair correlation function, mark correlation function, broad-leaved Korean pine mixed forest

## Abstract

Investigating the spatial distributions and associations of tree populations provides better insights into the dynamics and processes that shape the forest community. Korean pine (*Pinus koraiensis*) is one of the most important tree species in broad-leaved Korean pine mixed forests (BKMFs), and little is known about the spatial point patterns of and associations between Korean pine and community-level woody species groups such as coniferous and deciduous trees in different developmental stages. This study investigated the spatial patterns of Korean pine (KP) trees and then analyzed how the spatial associations between KP trees and other tree species at the community level vary in different BKMFs. Extensive data collected from five relatively large sample plots, covering a substantial area within the natural distribution range of KP in northeastern China, were utilized. Uni- and bivariate pair correlation functions and mark correlation functions were applied to analyze spatial distribution patterns and spatial associations. The DBH (diameter at breast height) histogram of KP trees in northeastern China revealed that the regeneration process was very poor in the Changbai Mountain (CBS) plot, while the other four plots exhibited moderate or expanding population structures. KP trees were significantly aggregated at scales up to 10 m under the HPP null model, and the aggregation scales decreased with the increase in size classes. Positive or negative spatial associations were observed among different life stages of KP trees in different plots. The life history stages of the coniferous tree group showed positive spatial associations with KP saplings and juvenile trees at small scales, and spatial independence or negative correlations with larger KP trees at greater scales. All broad-leaved tree groups (canopy, middle, and understory layers) exhibited only slightly positive associations with KP trees at small scales, and dominant negative associations were observed at most scales. Our results demonstrate that mature KP trees have strong importance in the spatial patterns of KP populations, and site heterogeneity, limited seed dispersal, and interspecific competition characterize the spatial patterns of KP trees and community-level spatial associations with respect to KP trees, which can serve as a theoretical basis for the management and restoration of BKMFs in northeastern China.

## 1. Introduction

BKMFs are primary forests mainly distributed in northeastern Asia, including the Korean Peninsula, the Far East Region of Russia, Japan, and northeastern China [1,2]. They are well known for their relatively high productivity, unique species composition, and high biodiversity value compared with those of other temperate forests at the same latitude [3,4,5,6]. In recent decades, a large area of BKMF in northeastern China was transformed into secondary forests lacking KP trees due to deforestation for timber production, seed collection, and natural disturbances such as fires [5,7,8]. KP trees have great economic value due to their high-quality wood and nutrient-rich seeds, as well as their important ecological value as a key species for biodiversity conservation [7]. Therefore, understanding the population structure (population size, age structure, and spatial distribution) of KP trees and the spatial relationship between this species and other tree species is vital to restoring KP trees and maintaining the crucial ecological functions of BKMFs in northeastern China [9,10].

In general, the size class structure and spatial distribution patterns of trees are considered important features of population structure [11,12]. In particular, the spatial distribution of and association among trees in different life stages in a population can provide insights into underlying ecological processes, such as regeneration, succession, mortality, intraspecific interactions, and the relationship with the surrounding environment [13,14,15,16]. Likewise, the spatial association among different populations can also provide important information on forest community dynamics and the mechanism of biodiversity maintenance [17,18,19,20]. Spatial point pattern analysis, as a powerful tool for completing the aforementioned tasks, has received increasing attention from ecologists; thus, it has been widely applied to investigate the spatial distribution and association of plant communities [21,22,23,24,25]. This approach treats the position of each tree as a point defined by geographical coordinates and analyzes the point patterns and spatial associations at any scale by utilizing the spatial information to the maximum extent [22,26]. Typically, the spatial distribution of points is divided into three groups, random, clustered (aggregated or clumped), and regular (uniform or segregated), while the spatial association between two different types of points is categorized into attraction (positive correlation/association), repulsion (negative correlation/association), or independence (no correlation) [27,28]. These different univariate and bivariate spatial patterns can be generated by various factors and processes, such as environmental heterogeneity [28,29,30], dispersal limitation [22,31,32], shared seed dispersal [33,34], negative distance [35,36,37], density dependence [38,39,40], and disturbance history [1,41].

A variety of studies on the community structure and spatial distribution dynamics in BKMFs have mainly focused on a few major tree species, including Korean pine, with high importance values in the forest community [42,43,44,45]. However, these studies were mainly restricted to individual plots and investigated the spatial association among individual populations of those major tree species. Studies revealing the spatial relationship among KP tree populations, overall needle-leaved trees, and overall broad-leaved trees have not yet been conducted. Furthermore, most KP-related studies were carried out in the Changbai Mountain plot, and only a limited number of investigations have explored the spatial patterns and associations of KP trees in other plots of northeastern China [38,46,47]. It remains poorly understood how the spatial distribution of and association between KP trees and other tree species change with different latitudes of habitats. Even though trees are of the same species, they may exhibit distinctive spatial distribution patterns that are dependent on their habitat and life stage. Likewise, the spatial association between focal trees and conspecific/heterospecific trees varies among growth stages and habitats. For example, Zhang et al. (2012) found that the spatial distribution patterns of KP trees differed between natural and plantation forests, while Zhou et al. (2019) demonstrated that the spatial association between KP trees and other woody species varied depending on soil nutrient content [11,48]. Similarly, Gu et al. (2019) showed that the spatial interaction between KP trees and other species was influenced by the habitat type [49]. Hence, it is of great importance to understand how the spatial distribution of and spatial relationship of Korean pine trees and other coexisting tree species differ depending on the latitude as well as growth stages.

The overall objective of this study is to investigate the spatial distribution patterns of and intra- and interspecific associations among different life stages of KP trees and other tree species at the community level in northeastern China. Specifically, we address the following research questions: (1) Are there significant differences in the population structure of KP trees among the study regions? (2) What are the distribution patterns of and interspecific associations among different life stages of KP trees? (3) How do the spatial associations between Korean pine trees and other community-level coniferous and broad-leaved tree species groups vary among different life stages and research areas?

## 2. Results

### 2.1. Population Structure of Korean Pine Trees

The DBH structures of KP trees in five BKMFs in northeastern China are illustrated in Figure 1. KP trees in the CBS plot showed a unimodal- and bell-shaped distribution with a peak in the mature stage. In contrast, the ML and SS plots had exponentially descending distributions, and the LS and FL plots displayed bimodal shapes with the two highest peaks in the sapling and mature stages. The mean DBH values for the five plots ranked as LS > FL > CBS > SS > ML. The ML and SS plots included a large proportion of saplings but fewer mature trees compared with other plots. In the ML and SS plots, saplings accounted for the largest proportion, 53.57% and 42.44%, respectively, whereas the CBS plot had only three saplings. Meanwhile, mature trees in the CBS, LS, and FL plots accounted for more than 93%, 77.28% and 65.3% of the total KP tree numbers in the plots, respectively. The LS and FL plots also had relatively large numbers of sapling trees (8.34% and 18.33%, respectively) compared with that in the CBS plot (0.13%). The characteristics of the KP tree distributions in the five plots are given in detail in Table 1. These results reflect significantly different population structures of the KP trees in the five plots.

### 2.2. Spatial Distribution of and Spatial Association among Life Stages of KP Trees

#### 2.2.1. Point Pattern Analysis of KP Trees

The point distribution of the four life stages of KP trees in the plots are illustrated in Figure 2.

The spatial patterns of the KP trees in different life stages as well as those of all KP trees were then analyzed using a univariate pair correlation function (g_11_(r) function) under a CSR null model (Figure 3a). All KP trees, saplings, and mature trees in the five plots displayed clustered patterns at all scales, except for the random distribution of all trees and mature trees in the LS plot at scales of 30–40 m. The juveniles in the CBS plot showed aggregation at the scale ranges of 0–10, 12–16, and 27–31 m, while juveniles in the LS plot exhibited a random distribution at all scales except in the 10–16 m range. In addition, juveniles in the FL and SS plots displayed aggregated patterns at the 0–35 m and 0–31 m scale ranges, respectively. For large trees, the CBS plot showed aggregation at all scales, and the LS plot exhibited a random distribution pattern at most scales, while the FL plot showed a random distribution pattern at discontinuous scales of 0–9, 17–20, and 30–37 m and an aggregation pattern at the remaining scales. Taken together, the CSR null model-based prediction of the spatial distribution patterns of KP trees revealed that KP saplings and mature trees had clustered (aggregation) patterns at most scales, while juveniles and large trees showed hybridized patterns of aggregation and random distributions across the range of scales.

Compared with the CSR null model, the degree of aggregation of KP trees in different life stages at all scales decreased in the HPP model (Figure 3b). In particular, the CBS, LS, and SS plots exhibited similar distribution patterns of KP trees, in which the trees were aggregated at small scales and then shifted to be randomly distributed or uniformly distributed as the scale increased. The KP trees in these plots exhibited aggregated patterns at the 0 to 10 m scale and random or uniform distribution patterns at larger scales. The distribution patterns of the KP trees were also dependent on the life stage. For the sapling and juvenile stages, the FL and ML plots showed aggregation patterns at most scales, while the CBS, LS, and SS plots exhibited only small proportions of aggregation patterns at the 0–10 m scale. For the mature stages, the ML and FL plots revealed aggregation patterns at most scales, while the CBS, LS, and SS plots had hybrid patterns of aggregated, random, and uniform distributions. Large KP trees mostly showed random distribution patterns; all of the LS plot trees showed a random distribution, while the CBS and FL plots additionally had certain proportions of regular and aggregated patterns, respectively. Overall, the HPP null model further considered the heterogeneity of the plot and improved the prediction of the spatial distribution patterns of KP trees.

#### 2.2.2. Intraspecific Association of KP Trees

We employed the bivariate pair correlation function (g_12_(r) function) to analyze the spatial associations within the population of KP trees in different developmental phases (Figure 4). In all plots, saplings exhibited a positive association with juveniles at almost all scales, with the exception of the LS plot, in which saplings and juveniles had a positive association at the scale of 0–7 m and a negative association at the scale of 11–40 m. In addition, the spatial relation of saplings and mature trees exhibited a negative association in the LS, FL, and SS plots and a positive association in the ML plot at all scales. A comparison of saplings and large trees showed a negative association at all scales in the LS and FL plots. The association between juveniles and mature trees in all the plots exhibited similar distribution profiles. In detail, a positive association between juveniles and mature trees was found at all scales in all plots except the SS plot, in which the association was positive only at the 0–10 m scale. Further comparison of juveniles and large trees mostly revealed independence at most scales as well as an insignificant (negative) association at the 20–24 m scale in the CBS plot and positive associations at the 7–11 m and 32–40 m scales in the LS plot. In the FL plot, a positive association between juveniles and large trees was noted at the 0–10 m scale, whereas no association was noted at larger scales. The association between mature and large KP trees was negative at most scales in the CBS plot but neutral at most scales in the FL and LS plots.

We then applied the univariate mark correlation function to assess the association between DBH sizes and the distance between point pairs (Figure 5). A total of five DBH classes, including “all KP trees”, “saplings”, “juveniles”, “mature trees”, and “large trees”, were used. For all KP trees, it was predominantly found that the trees had negative correlations at small scales. For instance, negative correlations for all KP trees were found at the 0–7 m, 0–9 m, and 0–4 m scales in the LS, FL, and SS plots, respectively. The CBS and ML plots also revealed stronger negative correlations at the 0–7 m scale than at the remaining scales. Saplings in the ML plot presented a strong positive correlation, while there was independence in other plots. Juveniles and overmature trees in all plots also showed independence at most scales but were statistically insignificant (*p*-values over 0.05), except for those in the FL plot. Mature trees exhibited negative correlations at all scales in the CBS, ML, and FL plots but only negative correlations at the 0–9 m scale in the LS plot. The SS plot presented independence at the 0–9 m scale and a positive correlation at the 9–37 m scale. Taken together, the results of the univariate mark correlation function analysis suggest that possible competition (or spatial segregation) and spatial independence were predominant among different life stages of KP trees in all plots.

### 2.3. Interspecific Association between Korean Pine and Other Tree Species

We employed the bivariate pair correlation function (g_12_(r) function) again to analyze the spatial correlation between KP trees and other tree species in different life stages (Figure 6, Figure 7, Figure 8 and Figure 9). As shown in Figure 6, KP saplings showed attraction patterns with ND trees in different life stages at the scale of 0–3 m under the HPP null model, while KP juveniles and ND trees in different life stages were positively correlated at small scales, followed by no or negative correlations as the scales increased. Mature KP trees and ND trees presented independence or a small proportion of positive correlations at small scales and repulsion at most scales. The results indicated that KP saplings and juveniles were positively associated with ND trees at small or larger scales, while mature KP trees mostly displayed repulsion or independence with respect to ND trees at small scales (except some positive correlations at small scales in FL).

The interspecific association of KP and broad-leaved canopy layer trees (BD1) was different from that between KP and ND trees (Figure 7). In the ML and SS plots, KP and BD1 trees in different life stages showed spatial independence or negative correlations at most scales. In the LS plot, only KP saplings presented a positive association with BD1 trees in different life stages, whereas KP juveniles and mature trees showed independence or repulsion with respect to BD1 trees in different life stages; in the FL plot, mature KP trees exhibited a positive correlation with BD1 trees in the sapling and juvenile stages at the 0–20 m scale. The results indicate that KP trees and BD1 trees normally had negative correlations or were independent, except for the trees in the LS plot.

The spatial association between KP trees and middle-layer broad-leaved (BD2) tree species showed patterns similar to those between KP and BD1 (Figure 8). In all plots, repulsion patterns were predominantly found between KP trees and BD2 trees in all life stages at most scales, except for several life stages in the LS and SS plots. In the LS plot, KP saplings displayed a positive correlation at most scales with BD2 trees in all life stages, whereas KP juveniles and mature trees showed insignificant spatial independence and negative associations with BD2 trees. In contrast to the predominant negative correlations or lack of correlation, KP trees and BD2 trees in the mature stage showed a positive correlation at the 16–22, 26–29, 31–35, and 39–40 m scales in the SS plot. Overall, a positive correlation was predominantly found only between KP saplings and BD2 trees in all life stages in the LS plot as well as between KP and BD2 trees in the mature stage in the SS plot, while a negative correlation was mostly found in other cases.

Similarly, KP trees and the understory-layer broad-leaved (BD3) tree species group predominantly displayed negative correlations at most scales in the five plots except in a few cases (Figure 9). In the ML plot, KP and BD3 trees in all life stages displayed negative correlations at all scales, while the trees in the FL and SS plots also showed the largest proportions of negative correlations. In addition to the negative correlations, independence or a positive correlation was also found among trees in the FL, SS, LS, and CBS plots. In particular, a positive correlation between KP saplings and BD3 trees in all life stages was found at most scales in the LS plot. Moreover, KP juveniles and mature trees showed independence from BD3 trees at a relatively large proportion of scales, especially in the CBS, LS, and FL plots.

Taken together, these findings reveal that the spatial association between KP trees and other BD tree populations is mostly negative in the five plots, implying potential mutual inhibition or competition among the tree species. Furthermore, independence was the second most common type of spatial association, while there was a positive correlation in a few cases, illustrating much more complex interactions among tree populations in different life stages among various study areas.

## 3. Discussion

### 3.1. Population Dynamics and Spatial Patterns of KP Trees in Northeastern China

The DBH structure of a tree species provides information on the age structure, growth rate, and recruitment pattern of the population [14,49,50]. For example, a skewed size distribution towards smaller trees may indicate poor recruitment or disturbance, while a bimodal distribution may represent different age cohorts. In this study, we found different distributions of DBH classes among the study plots, suggesting that KP populations were in various developmental stages in different plots. The regeneration status in the CBS plot was extremely poor, which is in line with the outcomes of various previous studies [51,52]. Other scientists also found that there were no Korean pine trees under their own canopy and that there were seedlings but no saplings had survived [47]. This very low recruitment could be attributed to light limitation and conspecific negative density dependence (CNDD), as suggested by some researchers [53]. Light plays a crucial role in the regeneration of trees because it directly impacts the growth of seedlings and their ability to thrive in forest environments [54]. In general, *P. koraiensis* seedlings are known to be shade-tolerant [55,56], although some studies suggest that they exhibit mid-tolerance [51] or even intolerance to shades [51,57]. While KP seedlings can survive and experience prolonged periods of slow growth under dense forest canopies, their light requirement increases as they mature [42,58]. In specific life stages, shade-tolerant species may necessitate a substantial level of light to support their growth [1]. Previous research revealed that Korean pine seedlings at the ages of 3 and 5 years exhibit better growth when exposed to 30% to 60% of full light, while 7-year-old saplings grow best under full light [59]. The CBS plot is located in an uneven-aged mature forest with many dense-canopy trees; thus, unfavorable light conditions due to the closed canopy could increase the mortality of saplings and juveniles [47,51]. Meanwhile, the CNDD effect due to competition for nutrient resources and the influence of harmful organisms such as insects and pathogens could inhibit the growth of KP seedlings and further affect colonization and development into saplings in the CBS plot [40,54]. Other reasons for low recruitment might be poor seed availability due to predation by granivores [10,58] and the biological characteristics of KP as a K-selected species producing a small number of seeds [42,60]. In addition, as reported since the 1980s, a significant growth decline in younger KP trees was observed in northeastern China as a result of climate warming. In particular, the younger KP trees that were more susceptible to death experienced higher mortality rates, leading to an overall decrease in the number and density of these trees in this region [55]. Meanwhile, the DBH structures in the ML and SS plots presented a reversed “J” shape with many more saplings and juveniles compared to mature and large trees. This indicates ongoing regeneration of the KP populations in these plots. Similarly, KP trees in the LS and FL plots exhibited a stable population developmental phase, characterized by more mature and large trees than in the ML and SS plots. Although the numbers of saplings and juveniles were smaller than those in the ML and SS plots, they were much greater than those in the CBS plot, suggesting that they were also of moderate regeneration status. The relatively good regeneration status and stable size class structures might be attributable to the near-mature phase of these forests with environmental conditions such as canopy gaps providing sufficient light [1,52,57] and moderate competition for resources [27,29], which are favorable for the regeneration and establishment of KP seedlings and juveniles.

The spatial distribution pattern of tree populations can be influenced by different factors, such as biological characteristics, including reproduction strategy, seed dispersal ability, and interactions within or among species, and abiotic factors, including resource conditions, elevation, and topography [16,35,61,62]. Depending on various conditions, tree life stages can display different distribution patterns, such as clustering, randomness, or uniform distribution patterns. Among them, a clustered distribution is the predominant pattern in natural forests, and it is generally accepted that the degree of aggregation decreases as trees age and the observation scale increases [18,63,64]. In line with previous studies, we found that most KP trees and life stages displayed significant clustered patterns at the scale of 0–40 m [42,44,46], except for juveniles and large trees in LS, which presented insignificant random distributions (*p* = 0.06, *p* = 0.12, respectively). Aggregation at scales greater than 10 m is usually attributed to the heterogeneity of the environment, while aggregation at scales < 10 m can be considered the result of plant–plant interactions [38,65]. After eliminating the environmental heterogeneity using the HPP null model, the aggregation scales dramatically decreased in three plots (CBS, LS, and SS). Meanwhile, in the ML and FL plots, the same patterns were observed under both null models, representing positive departures in both cases. This indicated that the environmental heterogeneity in the ML and FL plots did not significantly affect the distribution patterns of KP trees. On the one hand, the small-scale aggregations of KP mature trees under the HPP null model can be ascribed to the limited seed dispersal ability of KP trees with large cones [10,42,60]. In addition to habitat heterogeneity and limited seed dispersal, the low recruitment rate of mature trees could result in a clustered pattern [66]. On the other hand, the random and uniform patterns that were observed for all KP and mature trees in the CBS and LS plots, as well as saplings in the LS plot, might have been generated by intraspecific competition for resources or self-thinning processes [38,45]. The large KP trees in the LS and FL plots represented random and regular distributions, which might be because larger trees require more nutrients, light, and water resources, leading to competition with each other [42]. To assess the relationship between the pair correlation function results and KP DBH sizes, the univariate mark correlation function Kmm(r) was employed (mutual stimulation or mutual inhibition). The results of the Kmm(r) function demonstrated that clustering detected at small scales (by the univariate pair correlation function) had a negative effect on the DBH sizes of KP trees in all plots (Figure 5). Interestingly, for both the univariate g_11_(r) function and Kmm(r) function, the pattern of all KP trees was similar to that of mature trees, even in those plots (ML and SS) where the relative abundance of adults was the lowest. This indicates the strong importance that mature trees have in the spatial pattern of KP populations.

### 3.2. The Intraspecific Association of KP Trees and Interspecific Association with Other Tree Species at the Community Level

Intraspecific spatial association provides insights into the underlying patterns and processes of population dynamics, distributions, and interactions among individuals, which vary with the life stage. In general, positive correlations are dominantly produced among younger life stages, while negative associations and independence are mainly observed in later stages [67,68]. A strong positive association between KP saplings and KP juveniles was observed in our study, in accordance with the findings of previous studies [69]. This might be due to forest environmental heterogeneity, such as gaps [1], in which saplings and juveniles grow and develop without competition. Meanwhile, the negative correlation between sapling and juveniles in LS at r > 10 m scales could be explained by the outcome of the seed dispersal and high light availability in the gap. All saplings except for those in the ML plot showed a negative association with mature and large trees [70,71]. This seems unlikely because KP saplings could present aggregation with their parent trees due to their limited seed dispersal and shade tolerance. However, understory woody shrubs with a higher regeneration ability might encroach on the ecological niche of KP saplings so that the saplings and seedlings of KP trees are distributed under the canopy of mature KP trees with sparse understory shrub vegetation [72]. In addition, the Janzen–Connell hypothesis could provide a reasonable explanation for the spatial repulsion between saplings and mature trees in BKMFs [43,73,74,75]. For the ML plot showing strong aggregation of saplings and mature trees, it is likely that limited seed dispersal of KP trees, as well as environmental heterogeneity, resulted in aggregation. Although aggregation of saplings and mature trees was found in ML, mutual inhibition was observed in the mark correlation function results, as mentioned above. The positive associations between juveniles and mature trees are in accordance with the previous result of strong aggregation of lower height classes of KP trees around their parent trees [42]. The negative correlation between mature and large KP trees in CBS can be ascribed to interspecific competition for light and nutrient resources [76].

The spatial association between KP trees and other tree species varied with the different habitats, size classes, and observation scales. For all study plots, the interspecific associations between KP and ND trees were characterized by attraction at small scales among small-size class pairs and repulsion (or random at some scales) among larger class pairs at increasing scales. ND tree species are coniferous species like KP, and they share similar ecological characteristics with KP trees. They are representative climax trees in BKMFs with shade tolerance that allows them to survive under canopy trees [74]. Due to ecological niche overlapping, it seems that they are less likely to coexist with each other, and interspecific competition between these species might occur at small scales. However, our results revealed that KP saplings and ND saplings, KP saplings and ND juveniles, KP juveniles and ND saplings, and KP juveniles and ND juveniles displayed significant positive associations at the scale of 3–10 m. This suggests that lower height classes of KP and ND trees do not experience interspecific competition and can co-occur spatially, especially in gaps. Several studies showed that *Abies nephrolepis* exhibited a significant positive correlation with KP trees at the scale of 0–10 m [46]. A possible reason for this is related to the biological characteristics of coniferous species. Due to their poor seed dispersal and shade tolerance, KP and other coniferous trees might thrive together under the canopy of broad-leaved trees without enhanced competition [17]. The attraction between KP saplings and mature ND, and between KP juveniles and mature ND at small scales can be attributed to favorable conditions for the survival and growth of younger KP trees under mature ND trees [42]. However, mature KP trees tended to be negatively associated with ND saplings at most scales (except for a very small proportion of independence or a positive correlation at some scales). Similar results were found in other studies detecting a negative association or no association between mature KP trees and *Picea jezoensis* [77,78]. For both KP and ND mature trees, repulsion was dominant, indicating strong intraspecific competition for light and nutrients in later stages [67,79].

In our results, negative spatial associations were dominant for KP and BD species regardless of the size class in all plots (Figure 7, Figure 8 and Figure 9). For all BD life forms, the spatial relation with KP among small size classes was positive or neutral at small scales, while the spatial association among larger size classes was negative at most scales. It was unexpected that the spatial association between KP and all BD life forms in the ML and SS plots accounted for a large proportion of the negative associations across all scales and among all life stages (except the spatial association between KP and BD2). This suggests that KP trees undergo strong interspecific competition with all life forms of BD trees in these plots due to spatial heterogeneity, and light and nutrient resource requirements. The positive association of KP saplings and mature BD1 trees, and KP juveniles and mature BD1 trees at small scales indicated that broad-leaved canopy trees provide a suitable microenvironment for the establishment and growth of small KP size classes [42]. Likewise, in the FL plot, mature KP trees allowed BD1 saplings and juveniles to grow under their canopy. A variety of species pairs displayed different association results, including slight positive associations at dispersed scales, no associations, and mostly negative correlations depending on the life stage and study plot, indicating largely pronounced interspecific competition. Overall, it appears that KP trees, the dominant species in broad-leaved KP mixed forests, showed a negative spatial association with or spatial independence from other tree species, since the available resources for all community species are restricted [80].

In this study, we aimed to assess the spatial relationship between KP trees and other tree species at the community level, which is different from previous studies that merely analyzed two individual populations in relation to each other. However, community-level spatial analysis could fail to detect any effect of individual populations because populations with more individuals contribute more than other populations with fewer individuals [38]. Determination of spatial distributions and associations within the forest community should take into account a variety of factors, such as availability of resources, topography, climatic conditions, disturbance history, land development history, functional traits of plants, phytochemical interactions (allelopathy), etc., to capture a complete picture of the real underlying ecological process in the plant community [17,58,81,82,83].

## 4. Materials and Methods

### 4.1. Study Area

Five typical BKMFs along the latitudinal gradient in northeastern China (within a range of 40.8° N to 49.0° N and 134.5° E to 124.0° E) were selected as study sites [2,4]. Changbai Mountain National Nature Reserve (CBS) is located in Jilin Province, and the other four plots, namely, Muling National Nature Reserve (ML), Liangshui National Nature Reserve (LS), Fenglin National Nature Reserve (FL), and Shengshan National Nature Reserve (SS), are located in Heilongjiang Province (Figure 10).

These areas are characterized by a monsoon climate with relatively long, cold winters and warm, rainy summers. The coniferous tree species include *Abies nephrolepis*, *Abies holophylla*, *Larix olgensis*, *Picea jezoensis*, *Picea koraiensis*, and *Taxus cuspidata*. Common broad-leaved woody species are *Acer mono*, *Acer tegmentosum*, *Betula costata*, *Betula platyphylla*, *Juglans mandshurica*, *Populus ussuriensis, Quercus mongolica*, *Tilia amurensis*, *Ulmus japonica*, *Corylus mandshurica*, *Syringa reticulata* var. *amurensis*, and *Acanthopanax senticosus*. The general characteristics of the study areas are shown in Table 2.

### 4.2. Field Data Collection

Referring to the technical specifications of the BCI (Barro Colorado Island in Panama) 50 ha plot, all the plots were divided into quadrats of 20 m × 20 m; then, they were further divided into subplots of 5 m × 5 m. All trees at least 1 cm in DBH were identified to the species level (marked with plastic tags), and their DBH (with a diameter tape), geographical coordinates (using a GPS device), and other information (i.e., dead or alive) were recorded in all censuses following a standard field protocol [84].

In order to analyze the spatial point patterns of and associations among different developmental stages of tree species, the broad-leaved trees were divided into three growth forms according to the attainable maximum height, namely, canopy layer (CL ≥ 20 m), middle layer (MDL ≥ 5 m), and understory layer (UL ≤ 5 m). All the coniferous trees, including KP, belong to canopy layer, since these tree species can grow up to the maximum height of 30~40 m. Each growth form was then divided into different size classes (surrogate for life history stages) according to DBH. For the spatial distribution and intraspecific association analysis of KP trees, we defined four life history stages of KP: sapling (7.5 cm ≥ DBH), juvenile (15 cm ≥ DBH ≥ 7.5 cm), mature (70 cm ≥ DBH ≥ 15 cm), and large trees (DBH ≥ 70 cm). For the spatial association between KP trees and other tree species, the life history stages of needle-leaved trees (ND) and the canopy layer of broad-leaved trees (BD1) were divided into three stages, sapling (7.5 cm ≤ DBH), juvenile (15 cm ≥ DBH ≥ 7.5 cm), and mature (DBH ≥15 cm), while the life history stages of middle layer broad-leaved trees (BD2) and understory-layer broad-leaved trees (BD3) were segregated into sapling (4 cm ≥ DBH), juvenile (8 cm ≥ DBH ≥ 4 cm), and mature (DBH ≥ 8 cm), and sapling (2 cm ≥ DBH), juvenile (3 cm ≥ DBH ≥ 2 cm) and mature (DBH ≥ 3 cm), respectively [38,73,75]. These classifications were consistently applied across all five plots. The KP saplings in the CBS plot and large trees in the ML and SS plots were not analyzed due to their very small numbers (fewer than 50) [27,28,48].

### 4.3. Data Analysis

Different summary statistics are applied in spatial point pattern analysis, and among them, Ripley’s K function is the most commonly used summary function [19,27,85]. The K(r) function can be defined as the mean number of points within distance r from the typical point, divided by the intensity (λ = n/A).
(1)$$K(r)=\frac{A}{n^{2}}\sum_{i=1}^{n}\sum_{j=1}^{n}\frac{I_{r}(d_{ij})}{W_{ij}}(i\neq j)$$
where r is the scale of the distance, A is the area of the sampling plot, n is the number of points in the plot, and d_ij_ is the distance between point i and point j. I_r_(d_ij_) is an indicator function, and when d_ij_ ≤ r, I_r_(d_ij_) is 1; otherwise, I_r_(d_ij_) is 0. W_ij_ is the weighting factor for edge correction [86]. The weighting factor is the proportion of the circumference of the circle with radius r, centered on point i and passing through point j, which falls within the study area.

The pair correlation function [22,26,87], the g(r) function, is derived from the K function, as shown in Equation (2).
(2)$$g\left(r\right)=\frac{dK\left(r\right)}{2\pi r\times dr}$$

Similar to the K(r) function, it calculates the number of points at distance r from an arbitrary point of the pattern, divided by the intensity of the pattern. The circle in the K(r) function is replaced by a ring in the g(r) function, which improves the g(r) function by eliminating the cumulative effect of the K(r) function and making it more sensitive to small-scale effects [22,88]. In this study, we employed uni- and bivariate g(r) functions to examine the spatial distribution patterns of and association among different trees. In detail, the univariate g_11_(r) function was used to analyze the spatial distribution patterns of different life stages of KP trees, while the bivariate g_12_(r) function was applied to assess both intra- and interspecific spatial associations between different size classes of KP and other tree species. If g(r) is greater than 1, it indicates that points are more likely to be found at distance r than if they were randomly distributed. If g(r) is less than 1, it indicates that points are less likely to be found at distance r than if they were randomly distributed. In other words, in univariate analysis, g_11_(r) > 1 denotes clustering, and g_11_(r) < 1 indicates a regular pattern. In bivariate analysis, g_12_(r) > 1 indicates attraction between two different types, and g_12_(r) < 1 indicates repulsion. g(r) = 1 means random (univariate) or independence (bivariate). Null models are used to test the significance of observed patterns by comparing them with patterns expected under a null hypothesis and inform researchers about the underlying mechanisms that may be driving the observed patterns. Complete spatial randomness (CSR) is the most frequently used null model in spatial pattern analysis and assumes that all points are independent and pattern intensity λ is constant at any location of the plot. Heterogeneous Poisson process (HPP) takes into account the heterogeneity of a habitat, in that the intensity varies with location and represents point-to-point interactions [21,22,85,89]. CSR and HPP null models (based on the Epanechnikov kernel with a bandwidth of h = 30 m and default r of spatial resolution) were adopted in the pair correlation function analysis.

In addition to the g(r) function, the mark correlation function was also employed in this study. The mark correlation function Kmm(r) measures the degree of association between the values of the marks attached to pairs of points as a function of the distance between them. When the mark is quantitative, this function computes the size product of two marks (f (m1, m2) = {m1 × m2}) normalized by the squared mean mark value taken over all pairs [86]. The null model of independent marking was applied to randomly shuffle the marks among all trees, assuming that the locations and DBH values of trees were independent of each other [57,82]. The interpretation is that if Kmm(r) < 1, mutual inhibition exists between two marks, while Kmm(r) > 1 denotes that point pairs at distance r have mutual aggregation, and if Kmm(r) = 1, the marks are independent of each other [13,62].

For all spatial pattern analyses, we performed 199 Monte Carlo simulations of the null model, and the 95% simulation envelope with the fifth highest and lowest values was constructed to test for significant departure of the empirical function [90]. We also applied a goodness-of-fit (GoF) test to evaluate significant departure from the null models [91]. All analyses were conducted using the “spatstat” [92] package of R software [93].

## 5. Conclusions

This study evaluated the spatial distribution of and intra- and interspecific spatial associations between KP trees and other tree species in BKMFs in northeastern China. The analysis of the size class structure revealed different developmental phases and population dynamics of KP populations in BKMFs. The KP population in old-growth BKMFs in northeastern China displayed poor regeneration performance, while middle-aged BKMFs showed a relatively good regeneration status of KP trees. The spatial distribution of KP trees displayed an aggregated pattern at small scales among small size classes, whereas later life stages showed a random distribution at large scales. Meanwhile, the intraspecific spatial association among different life stages of KP trees changed with scale, life form, and plot. In addition, KP populations exhibited slightly positive correlations with the coniferous tree population and broad-leaved species groups at small scales, and most interspecific relations were dominated by negative associations, with some spatial independence. A pair correlation function analysis under different null models revealed that habitat heterogeneity, seed dispersal, and density dependence could help shape the community structure and interspecific interactions in BKMFs. Since the spatial patterns are the comprehensive products of various factors, many other factors should be considered in further studies to uncover the mechanisms underlying the processes resulting in the similarities and dissimilarities in population structure, spatial distribution, and spatial association between KP and other community-level species groups in BKMFs in northeastern China.

## Figures and Tables

**Figure 1 plants-12-02906-f001:**
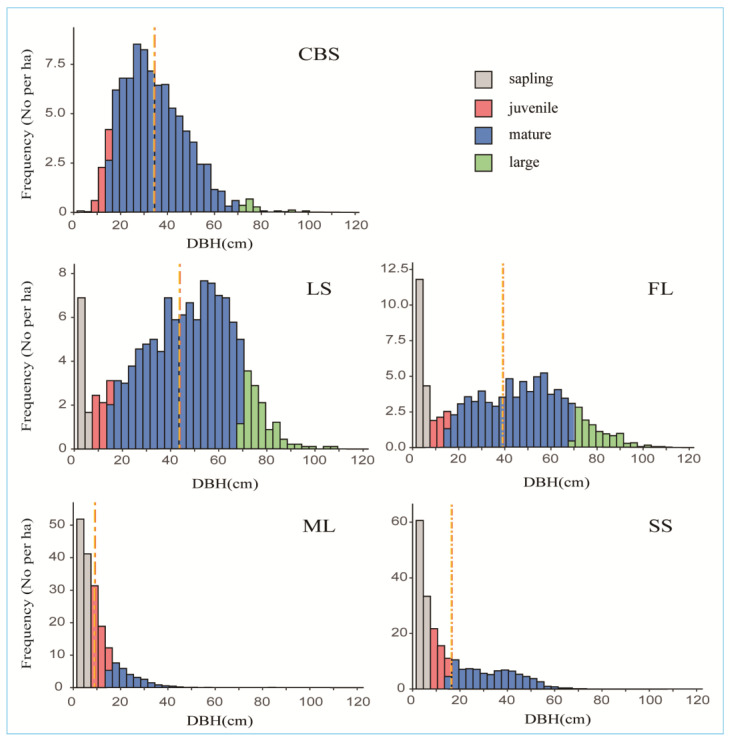
DBH histogram of different life stages of KP trees in the study areas. The yellow dotted line indicates the DBH mean, and different colors represent different DBH classes.

**Figure 2 plants-12-02906-f002:**
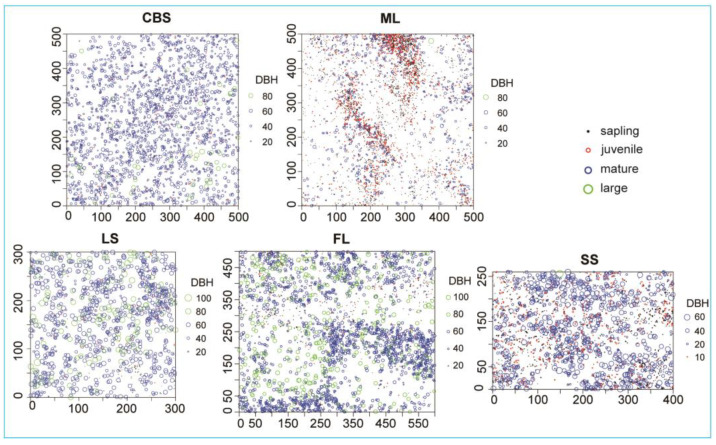
Point distribution patterns of different life stages of KP trees in 5 study plots. The size of the circles is proportional to the DBH of KP trees.

**Figure 3 plants-12-02906-f003:**
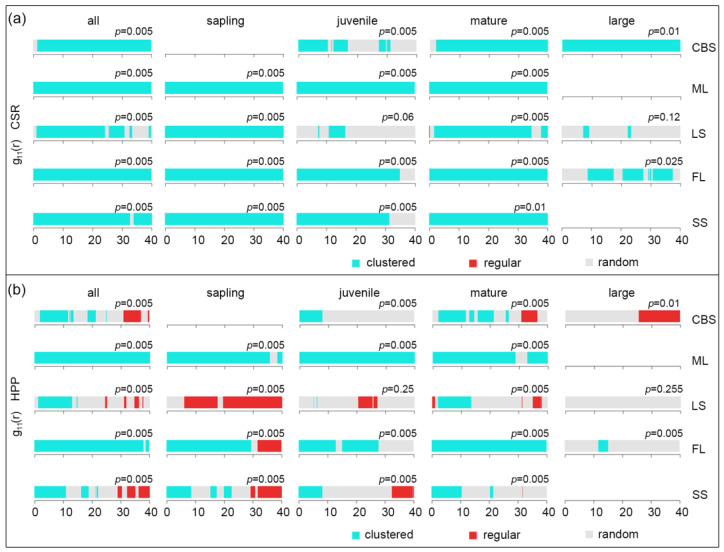
The results of the univariate pair correlation function under CSR and HPP null models. Panel (**a**) shows the g(r) function under CSR, while panel (**b**) depicts the g(r) function under HPP. The *x*-axis denotes scale r (m).

**Figure 4 plants-12-02906-f004:**
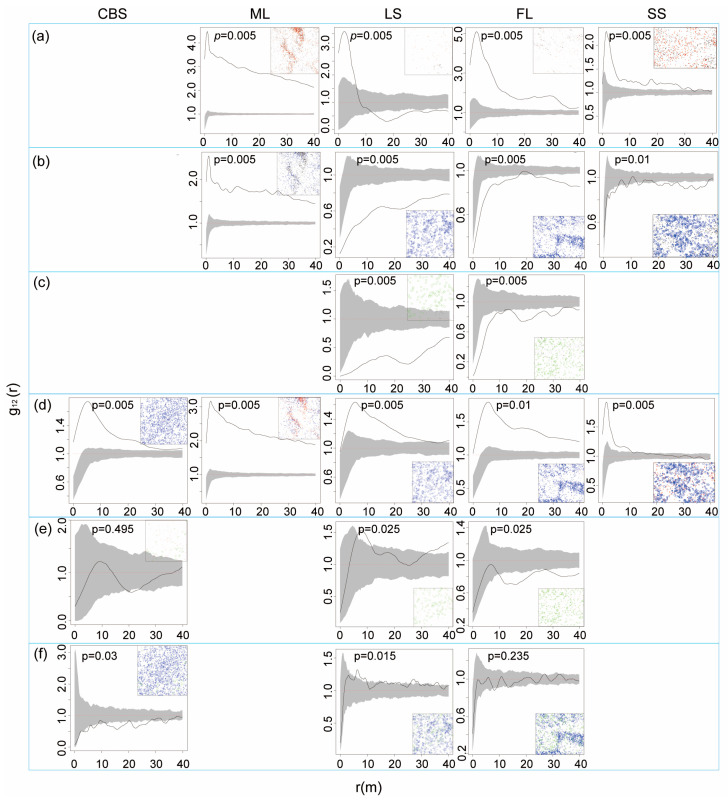
Intraspecific relationship between two different size classes. (**a**) Sapling vs. juvenile, (**b**) sapling vs. mature, (**c**) sapling vs. large tree, (**d**) juvenile vs. mature, (**e**) juvenile vs. large tree, and (**f**) mature vs. large tree. The *x*-axis denoted scale r (m), and the *y*-axis denotes values of the g_12_ function. The black line indicates the empirical function, and the grey envelope represents the 95% confidence interval. The red dotted line is the theoretical value. The black line above the upper envelope denotes a positive association; the black line below the envelope indicates a negative association; and the black line inside the envelope indicates no association.

**Figure 5 plants-12-02906-f005:**
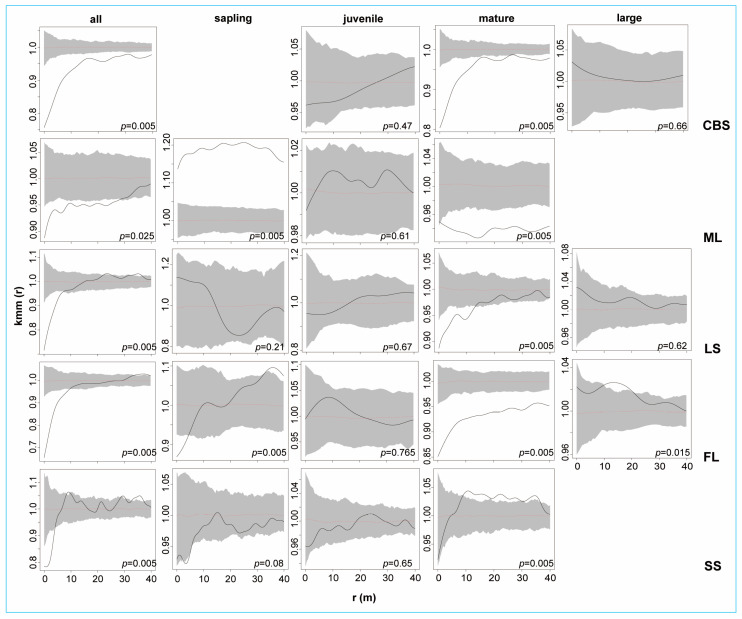
Mark correlation function results for different DBH classes of KP trees. The *x*-axis denoted scale r (m), and the *y*-axis denotes values of the Kmm function. The black line indicates the empirical function, and the grey envelope represents the 95% confidence interval. The red dotted line is the theoretical value. The black line above the envelope indicates a positive correlation; the black line under the envelope indicates a negative correlation; and the black line inside the envelope indicates independence.

**Figure 6 plants-12-02906-f006:**
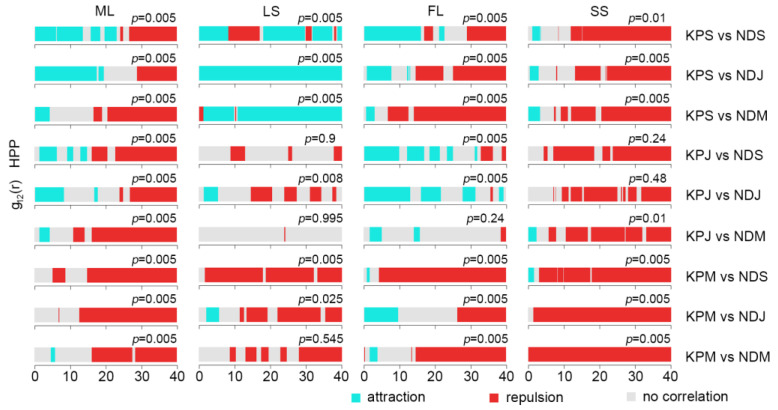
Bivariate pair correlation function results of different life stages of KP trees and ND species. KPS: Korean pine sapling; KPJ: Korean pine juvenile; KPM: mature Korean pine; NDS: coniferous tree sapling; NDJ: coniferous tree juvenile; NDM: mature coniferous tree. The *p*-value is the GoF test result.

**Figure 7 plants-12-02906-f007:**
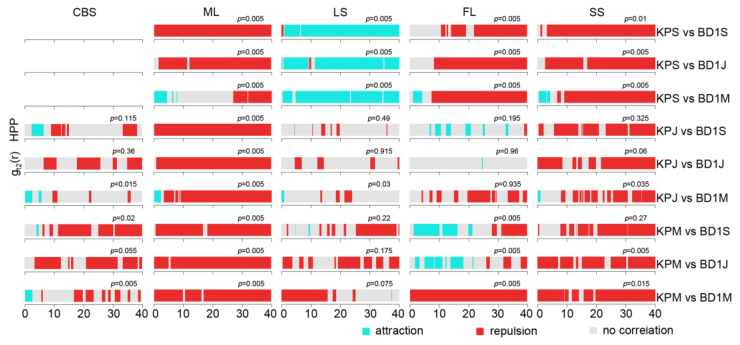
Pair correlation function results of different life stages of KP trees and BD1 tree species. KPS: KP sapling; KPJ: KP juvenile; KPM: mature KP trees; BD1S: BD1 sapling; BD1J: BD1 juveniles; BD1M: BD1 mature trees. The *p*-value is the GoF test result.

**Figure 8 plants-12-02906-f008:**
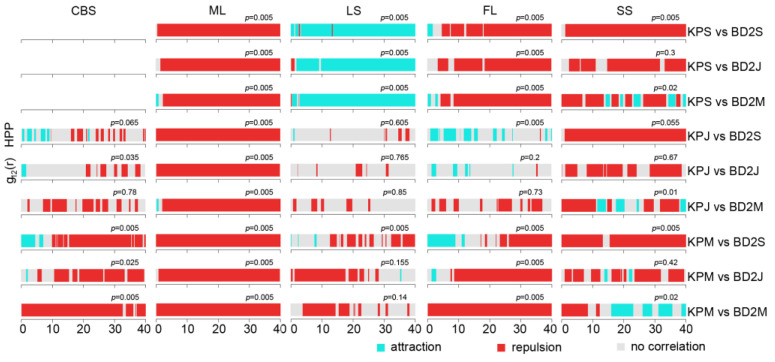
Pair correlation function results of different life stages of KP trees and BD2 tree species. KPS: KP sapling; KPJ: KP juvenile; KPM: mature KP trees; BD2S: BD2 sapling; BD2J: BD2 juveniles; BD2M: mature BD2 trees. The *p*-value is the GoF test result.

**Figure 9 plants-12-02906-f009:**
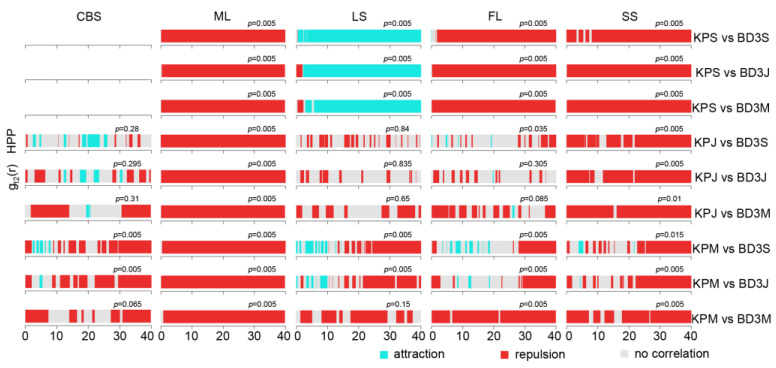
Pair correlation function results of different life stages of KP trees and BD3 tree species. KPS: KP sapling; KPJ: KP juvenile; KPM: mature KP trees; BD3S: BD3 sapling; BD3J: BD3 juveniles; BD3M: mature BD3 trees. The *p*-value is the GoF test result.

**Figure 10 plants-12-02906-f010:**
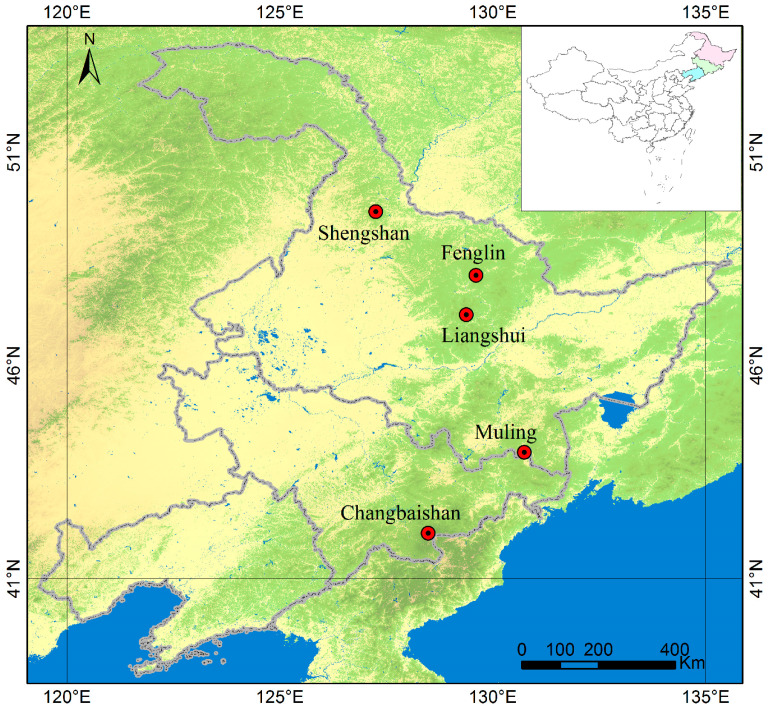
Location of the study areas.

**Table 1 plants-12-02906-t001:** The characteristics of stem density and DBH classes of KP trees in the study areas.

	CBS (25 ha)	ML (25 ha)	LS (9 ha)	FL (30 ha)	SS (10.4 ha)
Total No. of KP trees (n)	2286	4818	1175	3127	2347
Stem density (n/ha)	91.44	192.72	130.56	104.23	225.67
Saplings (n/ha)	0.12 (0.13%)	103.24 (53.57%)	10.89 (8.34%)	19.10 (18.33%)	95.77 (42.44%)
Juveniles (n/ha)	4.44 (4.86%)	57.20 (29.68%)	5.67 (4.34%)	5.23 (5.02%)	43.37 (19.22%)
Mature trees (n/ha)	85.16 (93.13%)	32.24 (16.73%)	100.89 (77.28%)	68.07 (65.30%)	86.44 (38.30%)
Large trees (n/ha)	1.72 (1.88%)	0.04 (0.02%)	13.11 (10.04%)	11.83 (11.35%)	0.1 (0.04%)
DBH mean (cm)	34.4	9.1	43.8	39.1	16.4
DBH max (cm)	98.4	84.1	109.0	111.6	71.3
DBH min (cm)	2.3	1.0	1.0	1.0	1.3

The values in parentheses represent the proportion of different life stages to the total KP trees in each plot.

**Table 2 plants-12-02906-t002:** General characteristics of the study sites.

Plot Name	CBS	ML	LS	FL	SS
Established year	2004	2013	2005	2009	2012
Inventory year	2014	2018	2015	2019	2022
Plot size (m^2^)	500 × 500	400 × 260	300 × 300	500 × 600	500 × 500
Latitude (N)	42.38°	43.95°	47.19°	48.08°	49°5
Longitude (E)	128.08°	130.07°	128.87°	129.12°	126°75
AE (m)	801.5	765	408	419	450
MAT (°C)	3.6	−2	−0.3	−0.5	−2
MAP (mm)	700	530	676	688	520

AE: average elevation; MAT: mean annual temperature; MAP: mean annual precipitation.

## Data Availability

The datasets used during the current study are available from the corresponding author upon reasonable request.

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
