# Peer review of "Spatial Distribution of Pinus koraiensis Trees and Community-Level Spatial Associations in Broad-Leaved Korean Pine Mixed Forests in Northeastern China"

_plants, 2023, doi:10.3390/plants12162906_

Round 1

Reviewer 1 Report

General opinion

This manuscript addresses a issues that have been addressed by researchers for several decades. Analyses of spatial patterns of plants to learn about the functioning of populations and communities has been conducted by numerous researchers over many decades.  However, few of these findings were referred to in the Introduction and Discussion sections; most of the cited papers were from local studies, and among the cited studies from other regions there is a quite big number of papers from tropical forests. On the other hand, information from temperate forests other than east-Asian are scarce. Of course, focusing upon the literature concerning the investigated species - Korean pine, in this case -  is understandable, but it has also its drawbacks, as some important problems concerning the relationship between pattern and process have been thoroughly studied in other tree species from the temperate zone (Eastern White Pine from North America is just one example).

The manuscript is arranged in a rather strange way, with Results and Discussion presented before the Material and Methods section; this is very inconvenient for the reader, especially that all the abbreviations, used throughout the text, are explained in the Material and Methods.

Data used for the statistical analyses came from five relatively large sample plots, scattered over the large area of north-eastern China within the natural distribution range of Korean Pine. All of the sample plots were located in national parks or in nature reserves, in well-preserved forest stands. In total the authors used data from the mapped stands of the total area of almost 100 ha; that needs to be stressed, as there are not so many papers dealing with spatial patterns of forest trees that had been based upon the analyses of such large data sets.

The classification of trees into "growth forms" and "life history stages" is complicated; the first classification was based upon tree heights, the second -  upon the tree DBH. What the authors presented are in fact size classes, as the relationship between diameter and age in trees is rather complex, and trees that are thicker are not necessarily older. It is not clear, what was the reason for defining the size class ranges differently for different plots. That part of the "Methods" section needs to be clarified.

The statistical analyses conducted by the authors are appropriate for answering the questions presented in the introduction. As the data base was large, the results are meaningful

In the "Results" section it would be very useful to have data about basal area per hectare (for Korean Pine, other species and total) for each of the study areas. Without that, interpreting of the diameter distribution is incomplete. The fact, that in almost all cases the spatial pattern of Korean pine trees was aggregated interesting; of course, analyses employing the HPP model indicate, that large part of it can be attributed to the heterogeneity of habitat conditions, but nevertheless the even in the HPP analyses the aggregated patterns are the most common. It is especially intriguing in case of the large trees (overmature trees, as the authors call them), because usually the largest trees in temperate forests are distributed randomly or even in a regular way.  

Discussion is not very convincing, focused upon explaining the results of this study with few references to other publications. The weakest part in the Discussion is combination of classifying Korean Pine as a "shade tolerant" species along with the information provided a few lines below that its seedlings grow better at "30% to 60% of full sunlight. This is not a feature of a shade-tolerant species. Including in the reference list examples from truly shade-tolerant species would allow the authors to discuss the shade tolerance issue in a more reasonable way.  

Specific remarks

Line 149: should be "aggregation"

Lines 209-201: the meaning of th8s sentence needs to be clarified

Lines 215-216: this is a strong statement, reaching far beyond the results presented in figure 5. Should be discussed more thoroughly

Lines 323-328: I cannot agree that the sample plots ML and SS represent a "good regeneration status". In case of really good regeneration, densities of saplings are many times higher than the densities of mature trees. Here, saplings constitute only between 43 and 53% of all trees measured in sample plots

Line 472: should be "ha", not "hm2"

Line 476: should be rather "broadleaved trees", not "broadleaved tree plants"

Line 481: the term "overmature" is not appropriate. Better call it "large trees"

Lines 484-488: the way of arrangement of species groups and size classes is very complicated; need to be presented and justified, why did you combine species groups and size classes in such a strange manner?

Line 554: "population group"; "population" is redundant here

the manuscript needs a moderate language editing

Reviewer 2 Report

the paper is very well written.
the statistical approach is original and well structured.
the study was carried out using correlation functions.
I don't think the sampling mode was presented

As is known, in general we have two competing hypotheses: a CSR/IRP process and a median value distribution process. Both cannot be rejected. This serves as a reminder that a hypothesis test cannot tell us whether a particular process is the process involved in generating our pattern of observed points; instead, it tells us that the hypothesis is one of many plausible processes.

on the other hand, the study is well conducted

As it is known, using g11(r) and L11(r) based on the null model of complete
spatial randomness (CSR), L11(r) (Lest) eliminates instability due to variance of the k-function and is not very sensitive to changes in the distribution pattern at a small scale, being better able to reflect changes on a large scale . In contrast, g11(r) is very sensitive to changes in the distribution pattern on a small scale.

The choices were well made

in the conclusions section, the method of selecting the hypothesis identified as the best should be more clearly explained.

Round 2

Reviewer 1 Report

The authors made some of the suggested changes, concerning terminology, cited literature and the issue of shade tolerance of the Korean Pine. However, several other suggestions (like presenting data about basal area of the analyzed stands) were not addreaased in the current version of the manuscript.

Author Response

Point 1: The authors made some of the suggested changes, concerning terminology, cited literature and the issue of shade tolerance of the Korean Pine. However, several other suggestions (like presenting data about basal area of the analyzed stands) were not addressed in the current version of the manuscript.

Response 1: We express our sincere gratitude for your thoughtful comments and suggestions. Your expertise and input have undoubtedly improved the quality of our article.

We carefully considered your suggestion to include data on basal area per hectare (for Korean Pine, other species, and total) in the "Results" section of our study. However, we regret to inform you that we do not possess and cannot present this specific data at the moment. We have already mentioned this in the previous version of response letter as follows:“However, at the time of this study, we did not collect specific data on basal area.”

We understand your concern that without the inclusion of basal area information, the interpretation of the diameter distribution might be deemed incomplete. However, we would like to respectfully emphasize that the analysis of the size structure of a given tree population can still be adequately performed by focusing on the diameter distribution alone. The diameter distribution provides valuable insights into the age structure, growth patterns, and overall health of the tree population under investigation. By analyzing the frequency and distribution of diameters, we can infer important aspects such as recruitment and overall population dynamics. Although basal area data would indeed complement our findings, we believe that the diameter distribution remains a crucial and informative metric for understanding the studied tree population.

Nevertheless, we acknowledge the importance of considering basal area data for a comprehensive analysis. In future studies, we will make efforts to obtain such data to further enhance the comprehensiveness of our research. We appreciate your suggestion and will duly address it in our future work.

We have also added a new sentence to support the conclusion. The sentence reads as “The KP population in old-growth BKMF of Northeastern China displayed a poor regeneration performance, while middle-aged BKMFs showed a relatively good regeneration of KP trees.” Please see line 567-569 in the revised manuscript.